# FISH–Flow Cytometry Reveals Microbiome-Wide Changes in Post-Translational Modification and Altered Microbial Abundance Among Children with Inflammatory Bowel Disease

**DOI:** 10.3390/pathogens13121102

**Published:** 2024-12-13

**Authors:** Mevlut Ulas, Seamus Hussey, Annemarie Broderick, Emer Fitzpatrick, Cara Dunne, Sarah Cooper, Anna Dominik, Billy Bourke

**Affiliations:** 1Department of Paediatrics, University College Dublin, CHI-Crumlin, D12 N512 Dublin, Ireland; ulasmevlt@gmail.com (M.U.); seamus.husssey@childrenshealthireland.ie (S.H.); annemarie.broderick@childrenshealthireland.ie (A.B.); emer.fitzpatrick@childrenshealthireland.ie (E.F.); 2National Children’s Research Center, CHI-Crumlin, D12 N512 Dublin, Ireland; sarah.cooper@childrenshealthireland.ie (S.C.); anna.dominik@childrenshealthireland.ie (A.D.); 3Department Gastroenterology, CHI-Crumlin, D12 N512 Dublin, Ireland; cara.dunne@childrenshealthireland.ie

**Keywords:** microbiome, post-translational modification, tyrosine phosphorylation, FISH–flow cytometry, inflammatory bowel disease, child

## Abstract

Metaproteomic analysis of microbiome post-translation modifications (PTMm) is challenging, and little is known about the effects of inflammation on the bacterial PTM landscape in IBD. Here, we adapted and optimised fluorescence in situ hybridisation–flow cytometry (FISH-FC) to study microbiome-wide tyrosine phosphorylation (p-Tyr) in children with and without inflammatory bowel disease (IBD). Microbial p-Tyr signal was significantly higher in children with IBD, compared to those without. *Faecalibacterium prausnitzii*, *Bacteroidota*, *Gammaproteobacteria* and *Bifidobacteria* tended to be more abundant in IBD than in non-IBD control children but there were only minor differences in p-Tyr among these bacterial communities in those with and without IBD. p-Tyr was significantly lower in non-IBD children older than 9 yrs compared with those less than 9 yrs, and the effect was seen in all four bacterial subgroups studied. The opposite trend was seen in patients with IBD. p-Tyr overall is higher in children with IBD but the effects of inflammation on p-Tyr vary according to the bacterial community. The overall microbiome p-Tyr signal changes with age in healthy children. FISH-FC can be used to study the microbiome-wide PTM landscape.

## 1. Introduction

The concept of intestinal dysbiosis is now a central tenet of the pathogenesis of inflammatory bowel disease (IBD). However, there has been a lack of consistency among studies examining the variation in the abundance of individual bacterial subgroups in the gastrointestinal tract of patients with IBD [1]. Indeed, whether the observed alterations in abundance are causal or secondary (or both) in the inflammatory process is still uncertain [2,3]. Studies mainly rely on metagenomic analysis, which, while highly discriminatory, may be prone to emphasising the dissimilarity between samples [4], potentially masking underlying bacterial community inter-relationships that may contribute to intestinal health and disease. On the other hand, the presence of a common shared core proteome with at least intra-individual stability [5,6], despite variability in metagenomic and metatranscriptomic analysis, raises the possibility that dysbiosis reflects microbiome-wide dysfunction at a level other than the altered abundance of specific groups of organisms.

Post-translational modifications (PTMs) are common in bacteria and have fundamental roles in prokaryotic cellular processes [7]. In the microbiome, it has been calculated that there may be as many as 240,000 different PTMs among microbial proteins [8]. Tyrosine phosphorylation is a particularly important bacterial PTM, and its role, especially in polysaccharide production and export, is well studied [9,10,11]. Bacterial PTMs are sensitive to changes in the microbial environment. For example, previously, we have shown that bacterial tyrosine phosphorylation is affected by the intestinal redox environment, with reactive oxygen species (ROS), especially H_2_O_2_, causing the depletion of capsular polysaccharide across a variety of intestinal organisms via the alteration of protein tyrosine phosphorylation [12]. The substantial oxidative stress including the generation of hypochlorous acid, peroxynitrite and hydroxyl radicals [13,14] intrinsic to the inflammatory process in IBD can be predicted to have widespread effects on bacterial PTMs, especially among mucosa-associated organisms.

PTM analysis of complex microbial ecology is still in its infancy. In a seminal human microbiota PTM study, Zhang et al. provided compelling evidence for altered microbiome lysine acetylation in children with Crohn’s disease (CD) [15]. However, the accurate identification of microbiome-wide PTM and the bioinformatic analysis required pose major challenges [8] and, to date, only a handful of metaproteomics-based PTM-microbiome studies have been published [15,16].

Flow cytometry (FC) has been used extensively in biomedical research and clinical settings. This powerful technique uses the light scattering properties of cells or the fluorescent emissions of probes to characterise target cells. The advantage of FC lies in its ability to separate subpopulations based on a characteristic(s) of interest and to quantify the relative differences within these subpopulations. The development of effective techniques for the disaggregation and separation of bacteria from environmental material coupled with fluorescent hybridisation staining (FISH-FC) has facilitated in-depth phylogenetic analysis of complex samples, including samples from human faeces [17]. FISH-FC offers the potential of high-throughput microbiome analysis without the complex (and costly) technical and analytic challenges associated with bacterial metaproteomics.

Here, we adapted flow cytometry methodology to examine global microbiome tyrosine phosphorylation in mucosa-associated organisms and showed strong evidence for the upregulation of bacterial tyrosine phosphorylation in IBD in children at the time of diagnosis and an unexpected alteration in the tyrosine phosphorylation status of bacterial faecal communities in children as they progress from early to late childhood.

## 2. Material and Methods

A comprehensive overview of the methodological framework adopted in this study, highlighting the processes involved, is illustrated in Figure 1.

### 2.1. Participants and Sample Collection

The DOCHAS study (Determinants and Outcomes in CHildren and AdolescentS with IBD) is a prospective inception cohort study of all new paediatric patients with IBD diagnosed in Ireland since 2012, which commenced following local ethical approval (GEN/193/11, CHI Crumlin Ethics Committee). Patients were recruited prior to diagnosis while they were treatment-naïve and followed prospectively until discharge to adult services. Comprehensive environmental, clinical, endoscopic, histologic, radiological, laboratory and treatment data were collected. Patients underwent full evaluation and phenotyping according to the Porto criteria and Paris classification [18,19]. Patients without IBD who underwent colonoscopy participated as “controls”.

Colonic lavage samples were collected, as described by Watt et al. [20], from children enrolled in DOCHAS. All samples were taken from children before diagnosis and treatment. Children with any underlying non-IBD medical diagnoses evident before or as a consequence of the endoscopy (e.g., polyposis syndromes), or those on medication, were excluded. Briefly, following standard picolax bowel preparation, 5–10 mL of colonic lavage material was aspirated from the recto-sigmoid using a suction trap, transferred to a separate tube, placed immediately on ice and transported to the laboratory for storage, extraction and analysis. Preliminary optimisation studies determined no substantial difference in microbial quantification or diversity between fresh samples extracted directly following endoscopy versus those extracted following immediate freezing at −80 °C.

### 2.2. Microbiome Extraction

The Nycodenz extraction step was conducted based on the method described by Hevia et al. [21], with specific adaptations for liquid samples. Briefly, colonic washings, initially stored at −80 °C, were gradually thawed on ice before commencing the extraction process. Each sample was homogenised for two minutes using a Qiagen homogeniser (TissueRuptor II, Qiagen, Hilden, Germany Catalog: 9002757). For microbiome extraction, 10 mL of the homogenate was carefully layered onto the top of a clean centrifuge tube (Beckman Coulter, Catalog: 344060, 14 × 95 mm), which already contained 3.5 mL of 80% *w*/*v* Nycodenz solution. The samples were then centrifuged at 4 °C and 8000 rpm for 40 min using an SW 40 Ti rotor (Beckman Coulter, Brea, CA, USA) to achieve density gradient separation. The top layer, holding dissolved particles, was carefully removed after the centrifugation process. The layers containing the extracted microbiota were collected by adding 8 mL of ice-cold 0.9% NaCl solution. After centrifugation at 3500× *g* for 7 min at 4 °C, the supernatant was discarded. This washing step was repeated twice, with each repetition involving the use of 10 mL of 0.9% NaCl buffer. The extracted microbiome samples were then resuspended with 3 mL of 0.9% NaCl solution and divided into aliquots before being frozen at −80 °C for further preservation.

### 2.3. Microbial Viability Assessment Using Propidium Monoazide

Nycodenz-extracted microbiota were analysed for viability using Propidium MonoAzide (PMA, Cat. No. 40019, Biotium, Fremont, CA, USA), a photoreactive dye that selectively targets and binds to DNA from non-viable cells. Analysis was performed as per manufacturer’s instructions. Briefly, aliquots of 400 µL of the extracted microbial community suspension were pipetted into clear microcentrifuge tubes containing 400 µL of 0.9% NaCI. As controls, live and dead bacterial cells (*E. coli* K12) were prepared. Live bacterial cultures were grown in LB media until the OD_600_ reached approximately 1. To generate dead cell controls, bacterial cells were heat-inactivated at 90 °C for 5 min. For Gram-negative bacteria, 200 µL of a 5X Enhancer solution was added to each tube, resulting in a final Enhancer concentration of 1X. Working in low light conditions, 2.5 µL of PMA stock was added to each tube for a final concentration of 50 µM. The tubes were incubated in the dark for 10 min at room temperature, gently rocked and completely covered with foil. After the dark incubation, the samples were exposed to light to cross-link PMA to DNA. A commercial halogen lamp (>600 W) was employed, with tubes placed on an ice block 20 cm from the light source and exposed for 5–15 min. The setup for the halogen lamp included a clear tray with aluminium foil to reflect light upwards, thereby ensuring even exposure. Following PMA treatment, microbial cells were pelleted by centrifugation at 5000× *g* for 10 min, and the supernatant was carefully removed without disturbing the pellet in order to proceed to the DNA extraction step.

### 2.4. DNA Extraction and PCR/qPCR

Microbiome samples were analysed for diversity using PCR targeting a range of typical faecal microbiota constituents. Briefly, genomic DNA was extracted from the PMA-treated and non-treated microbial cells using Microbiome DNA purification kit (Invitrogen, Cat. No: A29790, Waltham, MA, USA) according to the manufacturer’s instructions. Purified DNA was eluted with 100 µL elution buffer and stored at −20 °C until further use. Quantitative polymerase chain reaction (qPCR) was performed using primers targeting a selected genomic DNA target of interest for the microbial community (Table 1). The same volume of eluted DNA was used for each PCR reaction, ensuring consistent DNA input across samples. qPCR detection of the target genes was performed using 5 µL of gDNA, 1x SYBR Green PCR mix (Meridian Bioscience, Cincinnati, OH, USA) and 10 mM forward and reverse primers as stated in Table 2. Cycling conditions were 95 °C for 3 m followed by 40 cycles of 95 °C for 15 s and 60 °C for 30 s. Bact-8F and 802R bacterial 16S rRNA gene primers were employed as internal controls for quantitative polymerase chain reaction (qPCR). The determination of ΔCt involved subtracting the Ct values obtained from species-specific primers, as outlined in Table 1, from their respective reference Ct values. The resulting ΔCt values were then graphically represented in Figure 2B.

### 2.5. Immunoblotting

Protein was extracted from microbiota and cell lines by B-PER Complete cell lysis buffer (Cat. No: 89821, ThermoFisher Scientific, Waltham, MA, USA) containing protease and phosphatase inhibitors. Protein concentration was determined by a detergent-compatible (DC) protein assay kit (Bio-Rad, Watford, UK). Protein samples (30 μg) were resolved by sodium dodecyl sulphate polyacrylamide gel electrophoresis (SDS-PAGE) and transferred to Hyband LFP polyvinylidene fluoride (PVDF) membrane (Millipore, Livingston, UK). The membrane was blocked in 5% (*w*/*v*) non-fat milk powder in TBS-Tween (0.1% tween) for 1 h at room temperature. Primary antibody anti-β-actin (1:3000, Cat. No: 8457, Cell Signalling, Danvers, MA, USA) was added in 5% (*w*/*v*) non-fat milk in TBS-T overnight at 4 °C. The membrane was washed in TBS-T and incubated with HRP-conjugated secondary anti-rabbit antibody (1:5000, Cat. No: 7074, Cell Signalling, USA) in 3% non-fat milk for 1.5 h at RT. Membrane was visualised using the SuperSignal West Dura chemiluminescence method (ThermoFisher Scientific, USA).

### 2.6. Fluorescence In Situ Hybridisation

Pre-Fixation and Sample Preparation: Microbiome samples underwent filtration using a 5 μm PVDF filter (MERCK Millex-SV Filter Unit, Darmstadt, Germany, Catalog: SLSV025LS) prior to the fixation process. Following filtration, samples were quantified using a spectrophotometer (Agilent Cary 60 UV-Vis, Santa Clara, CA, USA) to establish a standardised cell density for subsequent procedures. The sample density was adjusted to approximately 1 × 10^10^ cells/mL. The FISH protocol outlined by Parsley et al. (2010) was employed with optimisation for microbial samples to accommodate both Gram-negative and Gram-positive bacteria. Briefly, samples were centrifuged at 12,000 rpm for 5 min, after which supernatants were discarded. Pellets were resuspended in 1 mL of fixative buffer (4% formaldehyde in 1X PBS, pH 7.4) and subjected to 3 h of incubation on a rotator at room temperature. After the incubation period, cells were centrifuged at 12,000 rpm for 5 min, and the supernatant was removed. Pellets were washed with 50% ethanol and centrifuged at 12,000 rpm for 5 min. This washing step was repeated with 80% and 95% ethanol. The final fixed cell pellets were left uncovered at room temperature to allow complete removal of residual ethanol. The fixed cells were subsequently exposed to 1 mL of lysozyme solution (10 μg/mL lysozyme in 1X PBS) for 1 h on ice to facilitate permeabilisation. After lysozyme treatment, the microbiome samples were centrifuged at 12,000 rpm for 5 min, washed twice with 1 mL of 1X PBS and centrifuged again using the same parameters. The washed cells were then resuspended in hybridization buffer, which consisted of filter-sterilised (0.2 μm) 5X SSC buffer, 20% formamide, 0.4% Tween-20 and 50 μg/mL Herring DNA. The suspension was equally distributed into separate microcentrifuge tubes, each with a final volume of 500 μL, corresponding to the specific FISH probes and controls utilised in the experimental workflow (Table 2). Following a 30 min incubation at 37 °C, fluorescently labelled oligonucleotide probes (Table 2) were added to the respective tubes at a concentration of 10 ng/μL. The mixtures were then incubated in the dark overnight at 50 °C. The subsequent day, after a centrifugation step (12,000 rpm, 5 min), cells were resuspended in 500 μL of 0.1X SSC buffer and subjected to a 15 min incubation at 37 °C. This washing process was repeated twice to eliminate non-specifically bound probe.

### 2.7. Intracellular Staining (Immunostaining)

Cell pellets were subjected to blocking by incubating them in a solution of 1% BSA in PBS for 1 h at room temperature with gentle rotation. Subsequent to the blocking stage, cells were carefully washed twice using 1 mL of PBS (at 12,000 rpm for 5 min). The cell pellets were then resuspended in 200 μL of PBS, containing either 3 μL of PE-conjugated anti-phosphotyrosine antibody (PY20 clone, Catalog: 309310, Biolegend, San Diego, CA, USA) or PE mouse IgG2b, k Isotype control antibody (MPC-11 clone, Catalog: 400312, Biolegend). This incubation was carried out at room temperature with gentle rotation for 3 h while maintaining a dark environment to minimise light exposure. Upon completion of the incubation period, the cells were washed twice using 1 mL of PBS each time. Subsequently, the cells were resuspended in PBS and prepared for flow cytometry analysis.

### 2.8. Flow Cytometry

Flow cytometry analyses were carried out using the BD LSRFortessa™ Cell Analyzer (BD Biosciences, Franklin Lakes, NJ, USA). This analyser is equipped with a Violet (405 nm, 50 mW), Blue (488 nm, 50 mW), Red (640 nm, 40 mW) and Yellow-Green (561 nm, XX) laser. The Yellow-Green laser was employed for forward angle scatter (FSC), side angle scatter (SSC) and intensity measurements using appropriate filters for detection of PE-conjugated PY20 antibody signal. The red laser was used to detect APC fluorochrome signal from Cy5-conjugated FISH probes. All bacterial analyses were performed at low flow rate settings (10 μL/min) and sensitivity of 500. A total of 10,000 events were recorded and stored in list mode files. Data were analysed using the BD FACSDiva Software from BD Biosciences, USA. The background fluorescence of *E. coli* K12 pure culture and a representative microbiome sample were assessed using negative controls (NonEUB338-Cy5 and Isotope control PE antibody). Bacterial cells/particles were gated, and counting was performed until 10,000 events were observed within the gated area.

The proportion of target cells hybridised with the Cy5-labelled probe (Table 2) having fluorescence intensities was calculated among 10,000 events. This proportion was then adjusted by subtracting the background proportion measured using PE-Isotope control NonEUB338-Cy5 to obtain an accurate value for the microbiome samples. The proportion of each subgroup (subgenus, subspecies) was presented as percentage of total 10,000 events. In the flow cytometry plots, y-axis (SSC-A) represents side scatter area, which indicates cell complexity and size, while the x-axis reflects fluorescence from either Cy5-conjugated FISH probe (APC) or PE-conjugated anti-tyrosine phosphorylation antibody. Higher fluorescence intensity on the x-axis indicates stronger binding to the respective probe or antibody.

### 2.9. Statistical Analysis

For the statistical analysis, the probability of a type 1 error was set at *a*  =  0.05. For all data sets, statistical analysis was performed by first analysing the data for normal distribution (Shapiro–Wilk or Kolmogorov–Smirnov test); subsequently, the data were analysed by Mann–Whitney U test. All analysis and graph representation were performed using *Python* (3.8.5). Data for bacterial abundance were presented as percentage (%), and for tyrosine phosphorylation were presented as Geometric Mean Fluorescence Intensity. Statistical details for each figure can be found in the figure legends. (* denotes statistical significance *p* < 0.05, ** denotes statistical significance *p* < 0.01, *** denotes statistical significance *p* < 0.001 and “*ns*” stands for not statistically significant *p* > 0.05.)

## 3. Results and Discussion

### 3.1. Optimisation and Validation of Microbiome FISH-FC

Preliminary optimisation experiments were conducted using pure *E. coli* cultures and stool samples from an adult volunteer. The viability was analysed using PMA. The results show high levels of viable organisms across representative species of the microbiota with no substantial differences between members of major phyla (Figure 2A). PCR using probes to target a wide range of faecal organisms verified that Nycodenz-extracted samples were representative of faecal microbiota (Figure 2B). Finally, in order to ensure that the p-Tyr signal was of bacterial rather than host origin, a subgroup of patient-derived Nycodenz-extracted samples were (i) analysed before and after filtration through a 0.5 µm pore membrane to exclude non-bacterial structures (Figure 2C) and (ii), using Western blot, probed with an anti-actin antibody to rule out an effect from human proteins adsorbed onto bacterial cells (Figure 2D). Both excluded a substantial contribution of human-derived phosphotyrosine-containing proteins within the samples.

These results show that the Nycodenz extraction technique isolates representative samples of stool microbial communities that are viable and appear to be unaffected by the presence of non-bacterial organic aggregates or protein complexes. These experiments provide confidence in the accuracy of our subsequent experiments.

### 3.2. Patient Population and Sampling

In total, forty-nine samples (twenty-nine male, twenty female) were included in the analysis, comprising twenty-five non-IBD controls and twenty-four with IBD (thirteen Crohn’s Disease (CD), eight Ulcerative Colitis (UC) and three Inflammatory Bowel Disease-Unclassified (IBD-U)). The median age for IBD and non-IBD children was 12.33 yrs (range 5.09–15.75 yrs) and 12.19 yrs (range 4.56–17.68 yrs), respectively. All children with IBD were confirmed to have active colonic inflammation at the time of sampling.

The overall tyrosine phosphorylation data are indicated in Figure 3A,B. The p-Tyr signal was significantly higher among IBD patients when compared to non-IBD controls (*p* = 0.044). In the subgroup analysis (Figure 3C), p-Tyr was higher among all IBD subtypes compared with non-IBD controls, although the difference in the medians for those with UC was marginal. Those with IBDU had higher p-Tyr levels than controls, approaching significance (*p* = 0.053), although the sample size for this group of children was small (n= 3).

In the context of our previous observation that host-derived H_2_O_2_ targets microbial pathogens to attenuate capsular polysaccharide production and virulence via decreased bacterial p-Tyr, we anticipated that the microbiota in patients with active IBD might show an attenuated p-Tyr signal. Therefore, the overall increase in eubacterial p-Tyr in patients with inflammatory disease (Figure 3A) was unexpected. The underlying mechanism remains a matter of speculation. However, it being higher in each IBD subtype supports the likelihood that the process is common to the inflammatory environment. For example, it is possible that the oxidative stress experienced by the microbiota during acute inflammation [14] results in the inactivation of bacterial tyrosine phosphatases [23], thereby altering overall microbiome phosphotyrosine homeostasis. Given the apparent ubiquity of phosphotyrosine signalling among prokaryotes [24,25,26], it can be expected that any process that changes bacterial p-Tyr is likely to be global and have widespread effects on bacterial physiology [27]. While tyrosine phosphorylation affects bacterial metabolism and DNA binding, the most widespread effect (and one highly relevant to faecal microbiota) is its modulation of exopolysaccharide production [24]. Altered p-Tyr, whether decreased or increased, can be expected to affect bacterial capsule and biofilm production with complex implications for bacterial communities and their interaction with the host.

### 3.3. Abundance and p-Tyr Signal Stratified by Bacterial Community

We used previously validated probes to target four bacterial communities, representing members of the mucosal microbiota implicated as important in health and disease [1,2] at a range of bacterial hierarchies: Bac303 selects for most *Bacteroidaceae* in addition to *Prevotellaceae* and *Porphyromonadaceae* families, broadly constituting the *Bacteroidota* phylum. Fprau645 targets *Faecalibacterium prausnitzii* and close relatives among the *Firmucutes*. Bif164 targets the *Bifidobacterium* genus, a group of organisms that is particularly important among the microbiota of children, and GAM42a selects for *Gammaproteobacteria*, a class of organisms identified as contributing to the dysbiosis associated with intestinal inflammation.

Significantly higher numbers of all four bacterial subgroups were detected in samples from patients with IBD compared with non-IBD patients (Figure 4A,B). Higher numbers of these four microbial groups were evident in children with all three IBD subclassifications (CD, UC and IBDU; Figure 4C). There was a particularly notable increase in *Gammaproteobacteria* in patients with CD (Figure 4C). The p-Tyr signal trended higher for *Bacteriodota* and *Gammaproteobacteria* in those with IBD; however, the effect was very small and statistically insignificant (Figure 5).

A number of metagenomics studies have previously identified attenuated *F. prausnitzii* abundance in patients with CD [1]. In our study, we observed a statistically significant increased abundance of *F. prausnitzii* in children with IBD (*p* < 0.001). Moreover, an increased prevalence of *F. prausnitzii* was observed in all three subtypes of IBD (Figure 4B). It is noteworthy that in two previous studies (one each of CD and UC) in which mucosal samples from children were used, as in our study, there also was increased *F. prausnitzii* abundance in IBD [28,29]. It is also interesting that previous metagenomics studies have generally found a lower abundance of *Bacteroidales/Bacteroidota* in patients with IBD [2,30,31], which is in contrast to our study, where significantly increased numbers were observed.

On the other hand, we observed a marked increase in *Gammaproteobacteria* among children with IBD, which corroborates existing data and concepts around the expansion of facultative anaerobic organisms under the increased oxygen conditions associated with inflammation [2,31,32]. The overall high prevalence of *Gammaproteobacteria* in our study compared with the results from many metagenomic studies in adults likely reflects the subject age group and mucosal origin of the communities we analysed [33].

Given the change in p-Tyr signal in IBD versus non-IBD children, we anticipated that the IBD p-Tyr signature might be reflected in one of the major subgroups of organisms analysed using FISH-FC. However, there were no convincing differences in the p-Tyr signal between those with and without IBD (Figure 5). These data suggest that an unknown representative(s) of the commensal microbiota is responsible for the overall upregulation of p-Tyr in IBD.

### 3.4. Effect of Age on p-Tyr

We noticed an apparent increase in the p-Tyr signal in younger patients. In a post hoc analysis using a cutoff age of 9 yrs (chosen as the halfway point for the cohort 0–18 yrs), we noted a substantially higher microbiome-wide p-Tyr signature in the younger children without IBD (medians/ranges; *p* = 0.024; Figure 6A,B). It is noteworthy that these age-related differences were not evident among those children with IBD (Figure 6A). In fact, the pattern was reversed, with a trend towards higher p-Tyr in the older group with IBD, although this result was not significant (*p* = 0.38)

There were similar numbers of patients over 9 yrs and a similar overall age distribution in IBD and non-IBD groups. Therefore, the lower p-Tyr signal in non-IBD patients overall largely accounted for the differences in those older (>9 yrs) children.

There were no major differences in the abundances of *F. prausnitzii, Bacteroidota, Bifidobacteria* or *Gammaproteobacteria* between the older and younger groups (Figure 6C,D). The p-Tyr signal for all four groups of organisms was lower in the >9 yr olds, compared with the <9 yr olds, in non-IBD children, reaching statistical significance for *Bifidobacteria* (Figure 6C). In those with IBD, there was a consistent but non-statistically significant trend towards higher p-Tyr signal for all four bacterial subgroups in the >9 yr olds (Figure 6D).

Although the major changes in the microbiota community structure in children occur in early infancy, a number of stool microbiome studies have documented substantial changes in microbial subgroups in mid-childhood [34,35,36]. The progressive fall in faecal calprotectin levels, a marker of intestinal inflammation, among healthy children [37] underlines the dynamic change in the intestinal milieu beyond the immediate period of infancy. In this context the age-dependent decrease in the p-Tyr signal in apparently healthy children is intriguing. The phosphotyrosine signal among eubacteria in children less than 9 yrs without IBD was statistically significantly higher than those over 9 yrs, and this effect was not observed in those with IBD (Figure 6A). In effect, the lower overall p-Tyr levels in non-IBD controls versus IBD patients were accounted for by the older non-IBD patient group. It is noteworthy that the fall off in the p-Tyr signal in non-IBD patients over 9 yrs was common to all four bacterial subgroups studied, suggesting an alteration in the intestinal milieu in childhood with global effects on mucosa-associated microbiota. This observation deserves further exploration in a larger cohort of children.

## 4. Summary and Conclusions

While the study of bacterial PTMs and our understanding of their fundamental roles in bacterial physiology and virulence have advanced substantially in the last 2 decades, many challenges remain. Even among the same bacterial species, the phosphorylation sites reported in phosphoproteome datasets have low overlap [38]. With the added complexities associated with high-diversity microbiome studies, including relatively low PTM abundance, lack of ability to discriminate between peptides from related species, very large database search requirements and the complexity of accurate bioinformatic analysis, the widespread application of PTM metaproteomic analysis is in its infancy [8]. Here, we demonstrate the utility of FISH-FC as an alternative approach to interrogate the tyrosine phosphoprotein signal and relative abundance of faecal microbiota from clinical samples. We show substantial and complex alteration in microbiome-wide p-Tyr signal in inflammatory disease and the intriguing likelihood of a global bacterial phosphotyrosine signal that changes with age in healthy children.

While our data using FISH-FC on the relative abundances of bacterial communities such as *F. prausnitzii* and *Bacteroidota* are at variance with some of the existing concepts around dysbiosis in IBD, the results are similar to those from previous analysis of mucosal samples in children. Taken together, our findings underline the importance of taking into account differences in age (child vs. adult), environmental niche (stool vs. mucosa) and technique (PCR vs. FC) when interpreting differences in the abundance of specific bacterial communities.

Our findings indicate that FISH-FC can be used to interrogate the PTM landscape of the microbiome in health and disease. We show that the p-Tyr signal is altered in IBD compared with controls, and the change is not consistent across bacterial communities. An apparent decrease in the p-Tyr signal in older children without IBD points to a dynamic alteration in the intestinal environment during maturation.

## Figures and Tables

**Figure 1 pathogens-13-01102-f001:**
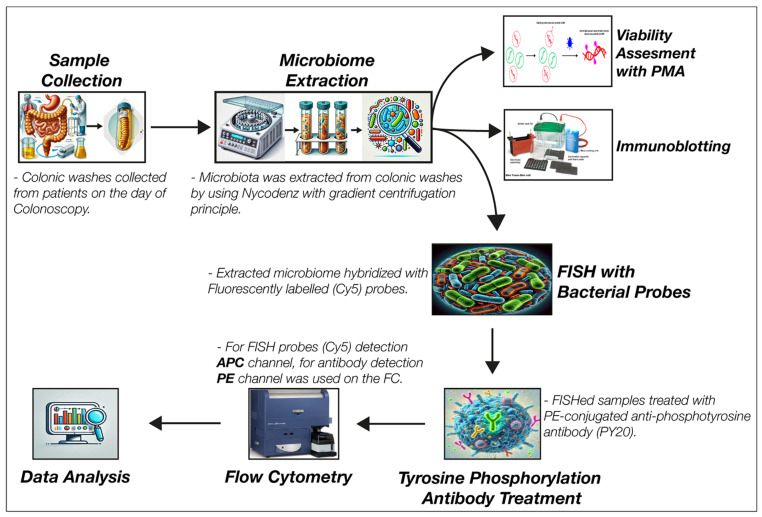
The research methodology applied in this study.

**Figure 2 pathogens-13-01102-f002:**
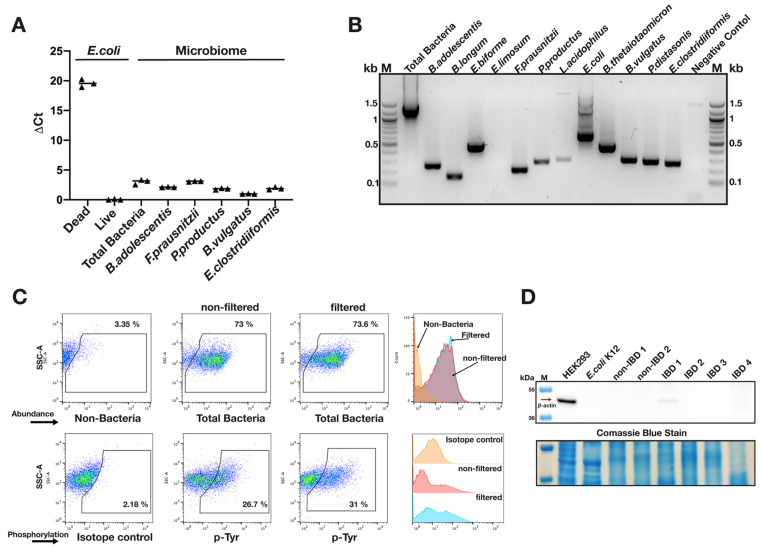
Optimisation and validation of FISH-FC technique. Nycodenz-derived microbiome samples maintain viability and diversity (**A**) Live-dead assay using PMA indicates high viability among species representing the main phyla of the faecal microbiota in a single representative Nycodenz-separated sample. *E. coli* K12 served as positive control. Viability of microbiome species is expressed as ΔCt value compared with heat-inactivated *E. coli* K12. (**B**) PCR amplification of a range of faecal bacteria confirms that Nycodenz-isolated microbiota maintain diversity. (**C**) FC histogram showing PY20 signal from a representative microbiome sample does not change substantially after filtration through a 5 µ pore filter to exclude non-bacterial cellular debris. (**D**) Western blot showing absence of reactivity for anti-actin antibody among nycodenz-extracted microbiota samples (2 non-IBD and 4 IBD samples) compared with human HEK cells indicates that extracted bacteria are not coated with host proteins.

**Figure 3 pathogens-13-01102-f003:**
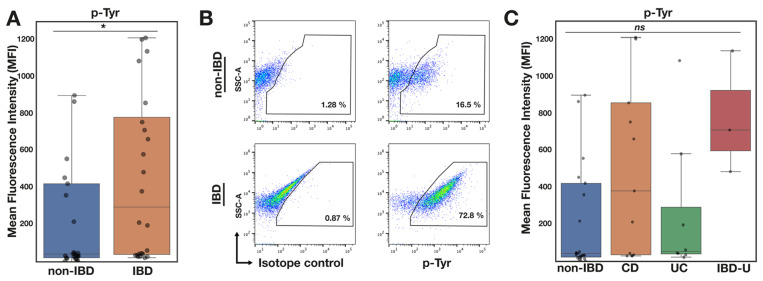
Higher p-Tyr signal among microbiota samples taken from children with IBD (n = 24) compared with non-IBD controls (n = 25). (**A**) Boxplot showing statistically significant higher median p-Tyr signal among IBD compared with control non-IBD samples. (**B**) Representative FC dot plot showing PY20 signal in non-IBD and IBD sample. (**C**) Boxplot showing PY20 signal is higher in IBD stratified according to IBD subtype (nCD = 13, nUC = 8, nIBD-U = 3). Data are means ± SEM, Mann–Whitney *U* test, * *p* < 0.05.

**Figure 4 pathogens-13-01102-f004:**
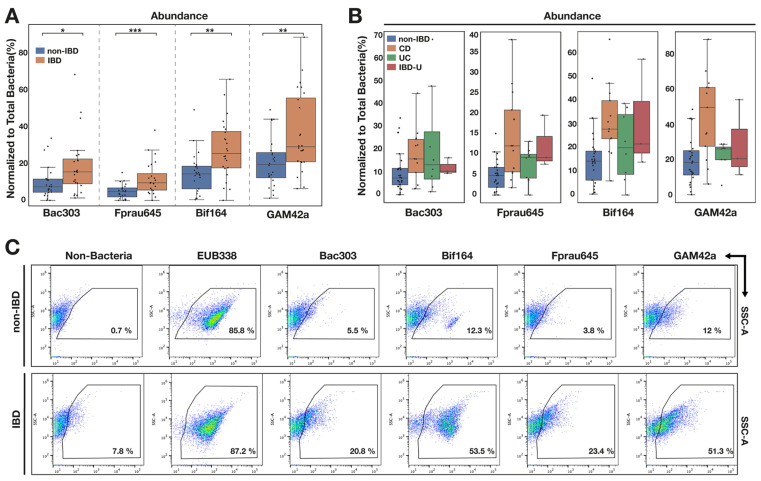
FISH-FC targeting *Bacteroidota*, *F. prausnitzii, Bifidobacteria* and *Gammaproteobacteria* indicates increases in abundance of all 4 bacterial subgroups in patients compared to those without IBD (n = 25), irrespective of the subtype of IBD. (**A**) Statistically significant increases in all 4 bacterial subgroups in IBD compared with non-IBD. (**B**) Boxplots of each bacterial subgroup stratified according to IBD type indicate higher abundances of *Bacteroidota, F. prausnitzii, Bifidobacteria* and *Gammaproteobacteria* in CD (n = 13), UC (n = 8) and IBD-U (n = 3). (**C**) Representative FC dot plots demonstrating increased abundance in IBD patients compared with non-IBD patients. Data are means ± SEM, Mann–Whitney *U* test, * *p* < 0.05, ** *p* < 0.01, *** *p* < 0.001.

**Figure 5 pathogens-13-01102-f005:**
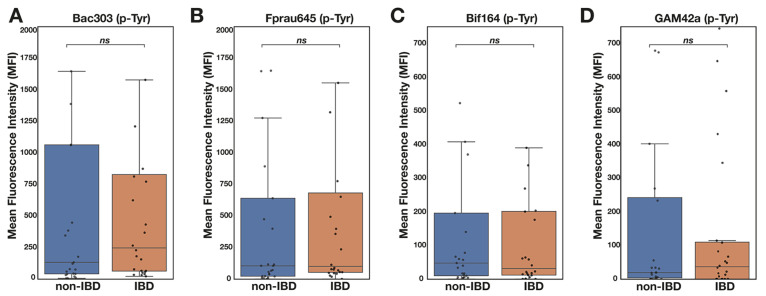
p-Tyr varies little in those with (n = 24) and without IBD (n = 25) among (**A**) *Bacteroidota,* (**B**) *F. prausnitzii,* (**C**) *Bifidobacteria* and (**D**) *Gammaproteobacteria* subgroups. p-Tyr signal was analysed among bacterial subgroups in those with and without IBD. There was a trend towards higher p-Tyr signal among *Bacteriodota* and *Gammaproteobacteria* but these differences did not reach statistical significance. Data are means ± SEM, Mann–Whitney *U* test.

**Figure 6 pathogens-13-01102-f006:**
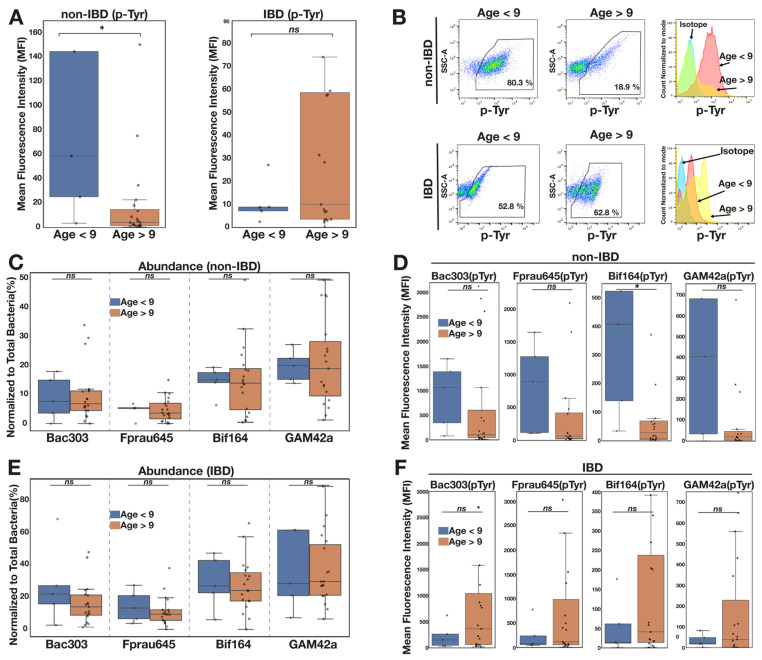
p-Tyr microbiome signal falls in healthy control children as they age. (**A**) In children without IBD over 9 yrs (n = 20), microbiota p-Tyr was significantly lower than those less than 9 yrs (n = 5) (* *p* = 0.024). However, the opposite trend is seen in children with IBD (n < 9 yrs = 5, n > 9 yrs = 19), where the p-Tyr signal increased, albeit without achieving statistical significance (*p* = 0.53) (**B**) Representative dot plot and histogram of p-Tyr in signal in individual patients younger and older than 9 yrs. (**C**) In children without IBD, no evidence of change in abundance of *Bacteroidota, F. prausnitzii, Bifidobacteria* and *Gammaproteobacteria* above and below 9 yrs, (**D**) while all 4 bacterial subgroups show evidence of reduced p-Tyr in children > 9 yrs, achieving significance for *Bifidobacteria* (* *p* = 0.029). (**E**) In contrast, while the bacterial subgroup abundance also did not vary by age in the IBD group, (**F**) the p-Tyr signal from each of the groups of bacteria studied tended to be elevated in IBD without achieving statistical significance. Data are means ± SEM, Mann–Whitney *U* test.

**Table 1 pathogens-13-01102-t001:** PCR primer pairs used in this study [22].

Target Group	Primer	Sequence (5′–3′)	Amplicon Size
Bacterial 16S rRNA	Bact-8F	AGAGTTTGATCCTGGCTCAG	794 bp
	802R	TACNVGGGTATCTAATCC	
*Bifidobacterium adolescentis*	BIA-1	GGAAAGATTCTATCGGTATGG	244 bp
	BIA-2	CTCCCAGTCAAAAGCGGTT	
*Bifidobacterium longum*	BIL-1	GTTCCCGACGGTCGTAGAG	153 bp
	BIL-2	GTGAGTTCCCGGCATAATCC	
*Eubacterium biforme*	EBI-1	GCTAAGGCCATGAACATGGA	463 bp
	EBI-2	GCCGTCCTCTTCTGTTCTC	
*Eubacterium limosum*	ELI-1	GGCTTGCTGGACAAATACTG	274 bp
	ELI-2	CTAGGCTCGTCAGAAGGATG	
*Faecalibacterium prausnitzii*	FPR-1	AGATGGCCTCGCGTCCGA	199 bp
	FPR-2	CCGAAGACCTTCTTCCTCC	
*Peptostreptococcus productus*	PSP-1	AACTCCGGTGGTATCAGATG	268 bp
	PSP-2	GGGGCTTCTGAGTCAGGTA	
*Lactobacillus acidophilus*	LAA-1	CATCCAGTGCAAACCTAAGAG	286 bp
	LAA-2	GATCCGCTTGCCTTCGCA	
*Escherichia coli*	ECO-1	GACCTCGGTTTAGTTCACAGA	585 bp
	ECO-2	CACACGCTGACGCTGACCA	
*Bacteroides thetaiotaomicron*	BT-1	GGCAGCATTTCAGTTTGCTTG	423 bp
	BT-2	GGTACATACAAAATTCCACACGT	
*Bacteroides vulgatus*	BV-1	GCATCATGAGTCCGCATGTTC	287 bp
	BV-2	TCCATACCCGACTTTATTCCTT	
*Parabacteroides distasonis*	BD-1	GTCGGACTAATACCGCATGAA	273 bp
	BD-2	TTACGATCCATAGAACCTTCAT	
*Enterocloster clostridiiformis*	CC-1	CCGCATGGCAGTGTGTGAAA	255 bp
	CC-2	CTGCTGATAGAGCTTTACATA	

**Table 2 pathogens-13-01102-t002:** FISH probes used in this study [17].

Probe	Sequence (5′–3′)	Target Group	Fluorescence
Eub338	GCT GCC TCC CGT AGG AGT	most Bacteria (16S rRNA)	Cyanine 5 (Cy5)
NonEub338	ACT CCT ACG GGA GGC AGC	control probe complementary to EUB338 (16s rRNA)- Negative control	Cyanine 5 (Cy5)
Bac303	CCA ATG TGG GGG ACC TT	most *Bacteroidaceae* and *Prevotellaceae*, some *Porphyromonadaceae* (16S rRNA)	Cyanine 5 (Cy5)
Bif164	CAT CCG GCA TTA CCA CCC	*Bifidobacterium* spp. (16S rRNA)	Cyanine 5 (Cy5)
Fprau645	CCT CTG CAC TAC TCA AGA AAA AC	*Faecalibacterium* (*Fusobacterium*) *prausnitzii* and relatives (16S rRNA)	Cyanine 5 (Cy5)
GAM42a	GCC TTC CCA CTT CGT TT	*Gammaproteobacteria* (23S rRNA)	Cyanine 5 (Cy5)

## Data Availability

All data are contained in the manuscript and Appendix A. Source data for all relevant figures is provided in the Appendix A.

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
