# Peer review of "FISH–Flow Cytometry Reveals Microbiome-Wide Changes in Post-Translational Modification and Altered Microbial Abundance Among Children with Inflammatory Bowel Disease"

_pathogens, 2024, doi:10.3390/pathogens13121102_

Round 1

Reviewer 1 Report

Comments and Suggestions for Authors

The authors addressed the interesting and important topic of changes in the microbiome in terms of post-translational modification and altered microbial counts in a group of children with inflammatory bowel disease using FISH-flow cytometry. The presented research is structured, logical and clear. The study was designed in an appropriate way. The authors described the analyzed (study and control) groups in detail. The study used modern, adequate molecular biology techniques. The results are presented in a clear, accessible and reliable manner. The results are not based on a large group of patients (n=24 with IBD), which is understandable given the age of the patients and the disease. However, the presented analysis offers interesting perspectives and may be a starting point for further research. Based on the results presented by the authors, it appears that the use of analyses using techniques such as FISH-FC can provide new information on changes in the microbiome in people with IBD, taking into account the patient's age and developmental period.

Author Response

We are grateful to the reviewer for the very positive comments on our manuscript.

Reviewer 2 Report

Comments and Suggestions for Authors

I have reviewed the article entitled "FISH – flow cytometry reveals microbiome-wide changes in post-translational modification and altered microbial abundance among children with inflammatory bowel disease," and I have found major flaws which need to be revised before further processing.

The whole article seems quite confusing because of the addition of irrelevant sections and sentences. The authors showed the introduction/material and methods in the results section. For example, in the first paragraph of section 3.3, Similar problems have repeatedly been shown in the abstract section, “FISH-20 FC can be used to study the microbiome-wide PTM landscape.

Original Fig 1B / Figure

There seem to be major flaws in the figure. For example, the negative control contains a band. The authors have included marker (M) with unknown band sizes (band size should be marked clearly to understand). The primer used for the bacterial 16S rRNA gene amplification can only amplify a 794 bp region (F8 & R802 = 794 bp size), not the entire gene. Why? The amplified region in the provided article may overlap with multiple bacteria. For example, Lactobacillus acidophilus and Escherichia coli had different amplification bands. Another issue is that frozen stool microaerophilic and frozen stool aerobic extractions should have different amplification patterns, which are not the same, as shown in the figure. It seems that the authors depicted the same figure in two distinct shapes.

Comments on the Quality of English Language

No major issue with language

Author Response

  1. The reviewer identified a sentence in the abstract section that was inadvertently misplaced and this has been moved to the end of the abstract in the revised manuscript.

  1. We have reviewed and optimized the flow of the manuscript in the revised submission. The first paragraph of section 3.3 we have left as per the original manuscript because it provides context to the results section immediately following. We feel that this approach is in keeping with the journal style and helps with the clarity of what is acknowledged by the editor and other reviewers as a complex set of results.

  1. The reviewer comments on original figure 1B:

- there is a faint band in the negative control which we assume to be non-specific binding. However it does not interfere with the overall interpretation of the gel.

-The molecular marker is present in the original figure and the main Ms figure 1B on the left top side of the figure.

-We used different amplification primers for the different organisms and, as seen in table 2, the sizes of the amplicons generally correspond with the predicted sizes.

-We noticed that the amplified product for eubacteria was larger than predicted and the cause of this is not clear. However, the result is not directly germane to the purpose or interpretation of the figure shown .

-We are not sure why the reviewer contends that the amplification patterns of the aerobic and microaerophilic extractions “should” be different. The fact that they are very similar is reassuring regarding that robustness off the samples analysed under different environmental conditions. We can absolutely confirm that two gels are different and do represent the two environmental conditions analysed – indeed careful scrutiny of the two gels’ banding patterns clearly corroborates that they cannot be the same.

Reviewer 3 Report

Comments and Suggestions for Authors

FISH – flow cytometry reveals microbiome-wide changes in post-translational modification and altered microbial abundance among children with inflammatory bowel disease

The experiment studied the adapted and optimized fluorescence in situ hybridization-flow cytometry (FISH-FC) to study microbiome wide tyrosine phosphorylation (p-Tyr) in children with and without inflammatory bowel disease (IBD). The results showed that microbial p-Tyr signal was significantly higher in children with, compared to those without, IBD. The authors concluded that FISH- 20 FC can be used to study the microbiome-wide PTM landscape. P-Tyr overall is higher in children with IBD but the effects of inflammation on p-Tyr vary according to the bacterial community.

Overall, the manuscript is very interesting, informative and well written.

Author Response

We thank the reviewer for the very positive comments on our submission

Reviewer 4 Report

Comments and Suggestions for Authors

The work is interesting and carried out well. The manuscript is well written.

Generally the letters in the figures are too small and the explanation of the color codes is unrecognizable.

Author Response

We thank the reviewer for the positive comments on our manuscript.

In the revised Ms we have addressed the issues around the figure legends and figure colour coding clarity, as suggested. 

Round 2

Reviewer 2 Report

Comments and Suggestions for Authors

The last sentence of the abstract could be “The overall microbiome p-Tyr signal changes with age in healthy children, suggesting that FISH-FC can be used to study the microbiome-wide PTM landscape”

The correct is “Materials and Methods not “Material and methods”, please revise carefully.

Comments on the Quality of English Language

The last sentence of the abstract could be “The overall microbiome p-Tyr signal changes with age in healthy children, suggesting that FISH-FC can be used to study the microbiome-wide PTM landscape”

The correct is “Materials and Methods not “Material and methods”